# Parathyroid near-infrared autofluorescence differently benefits depending on the surgeon's skill for preventing from hypoparathyroidism after total thyroidectomy: A systematic review and meta-analysis

Takeshi Takahashi[1☯‡*], Shalyn J. D. Sa[2☯‡], Ryohei Oya[3], Shusuke Ohshima[1], Jo Omata[1], Yusuke Yokoyama[1], Ryusuke Shodo[1], Yushi Ueki[1], Yukinori Takenaka[3], Hidenori Inohara[4], Arata Horii[1]

1 Department of Otolaryngology Head and Neck Surgery, Niigata University Graduate School of Medical and Dental Sciences, Niigata, Japan, 2 University of Leicester, Leicester, United Kingdom, 3 Department of Otorhinolaryngology Head and Neck Surgery, Osaka General Medical Center, Osaka, Japan, 4 Department of Otolaryngology Head and Neck Surgery, Osaka University Graduate School of Medicine, Suita, Japan

☯ These authors contributed equally to this work.
‡ These authors share first authorship on this work.
* tt_niigata@yahoo.co.jp

## Abstract

### Objective

To evaluate the role of parathyroid near-infrared autofluorescence in reducing the incidence of postoperative hypocalcemia and hypoparathyroidism after total thyroid-ectomy, and to determine which surgeons benefit most from parathyroid near-infrared autofluorescence use.

### Methods

A literature search was conducted in PubMed, Web of Science, and the Cochrane Library databases for English-language articles published from June 2011 to October 31, 2023. The inclusion criteria were studies conducted on patients who underwent total thyroidectomy for benign or malignant thyroid pathologies, comparing postoperative parathyroid function between parathyroid near-infrared autofluorescence techniques and conventional surgery with data on calcium and/or parathyroid hormone levels. The exclusion criteria included: reviews, letters, meta-analyses, case reports, animal experiments, or basic research. Of the initial 387 articles retrieved, we included 14. A meta-analysis was performed to calculate the pooled odds ratio and weighted mean deviation with a random-effects model. Main outcomes were Calcium and parathyroid hormone levels after total thyroidectomy with or without parathyroid near-infrared autofluorescence use.

**Data availability statement:** The data that support the findings of this study are available from the general secretary of our department, who keeps the data independently from the authors, and can be accessed upon request. Email: okayoko@med.niigata-u.ac.jp In addition, the concise dataset is available as S3 Table.

**Funding:** The author(s) received no specific funding for this work.

**Competing interests:** The authors have declared that no competing interests exist.

## Results

Fourteen studies were included in the meta-analysis. Pooled odds ratios of temporary and permanent hypocalcemia were 0.56 (95% confidence interval 0.43–0.72) and 0.61 (95% confidence interval 0.33–1.13), respectively. Meta-regression analysis revealed that near-infrared autofluorescence benefits surgeons with the high incidence of temporary hypocalcemia by naked eye surgery (≥15%) by reducing temporary hypocalcemia (p = 0.0091) and skillful surgeons by increasing the number of autotransplanted parathyroid glands (p = 0.0225).

## Conclusions

Parathyroid near-infrared autofluorescence has different benefits depending on the skill level of the surgeon.

## Introduction

Preservation of parathyroid function during thyroid surgery is important for preventing postoperative hypocalcemia (hypoCa). Permanent hypoparathyroidism (P-hypoPT) has been associated with a lifelong need for active vitamin D and/or calcium supplements; an increased risk of renal complications, seizures, and calcification of the basal ganglia [1,2]; and increased mortality [3]. Postoperative hypoPT is most often related to technical issues with thyroidectomy, such as inadvertent resection, direct injury, or devascularization of the parathyroid gland (PG). Although a systematic review in 2014 reported a 0–3% incidence of P-hypoPT after total thyroidectomy [4], recent epidemiological studies using national registries, insurance databases, and multicenter studies have shown a higher incidence rate of P-hypoPT, ranging from 11.2–15.0% [5–9], suggesting that the incidence of P-hypoPT is higher in reality. Therefore, it is necessary to explore novel techniques to assist in identifying PGs, rather than relying on the surgeons' experience and skills.

The use of exogenous contrast materials, such as methylene blue, 5-aminolevulinic acid, 99 m technetium-methoxybutylisonitrile, and carbon nanoparticle suspension injections, has been proposed to support the identification of normal PGs [10]; however, these methods are not widely practiced because of the need for drug administration and complicated procedures. In 2011, an innovative method based on near-infrared autofluorescence (NIRAF) of PGs was introduced [11]. This method can discriminate PGs with high accuracy (sensitivity: 98.5%; specificity: 97.2%) [12] and is reported to reduce the number of inadvertent PG resections during surgeries and increase the number of PG autotransplantations by applying it to resected specimens [12,13]. While there is improvement in the identification of PGs, the resultant parathyroid function with NIRAF use remains controversial. Although three systematic reviews and meta-analyses published in 2021[14–16] reported likely contributions to the prevention of temporary hypoCa (T-hypoCa), several issues remain unresolved. First, these reports were primarily cohort or case-control studies, and there were only two randomized controlled trials (RCTs)

[17,18]. Second, not only T-hypoCa but also P-hypoPT and permanent hypoCa (P-hypoCa) should be investigated. Third, it remains unclear which surgeons (experienced or novice) and disease types would benefit from NIRAF use for post-operative preservation of parathyroid function. Subsequently, four additional RCTs were reported in 2021–2023[19–22]; therefore, re-evaluation is warranted.

The aims of this meta-analysis were: 1) to determine whether intraoperative NIRAF use could significantly reduce the incidence of postoperative T/P-hypoCa and temporary hypoPT (T-hypoPT)/P-hypoPT compared with conventional search-ing methods relying on the naked eye (N-E) and 2) determine the surgeons or disease types (malignant or benign) that would benefit from NIRAF use.

## Materials and methods

### Search strategies

This systematic review and meta-analysis were conducted in accordance with the Primary Reporting Items for Systematic Reviews and Meta-Analyses Statement [23] (S1 Table) and the Cochrane Handbook for Systematic Reviews of Interven-tions [24]. We searched for relevant literature using PubMed, Web of Science, and the Cochrane Library published from June 2011 to October 31, 2023. The search protocol used was ((parathyroid), (parathyroid gland), or (parathyroid glands)) AND ((near-infrared autofluorescence), (autofluorescence), or (fluorescence)), and the reference lists of eligible studies were manually searched.

Initially, literature selection was based on the title and abstract. Subsequently, a full-text review of the relevant literature was performed and studies identified were independently assessed for inclusion/exclusion after full-text evaluation by two authors (TT and JO). Discrepancies were resolved by a consensus among the study teams. References were managed using Endnote 20 software (Clarivate Plc, Philadelphia, PA, USA). The search was conducted in November 2023.

### Inclusion criteria

The inclusion criteria were: (a) studies that included patients who underwent total thyroidectomy for thyroid disease (benign or malignant), (b) studies that compared postoperative parathyroid function between NIRAF techniques and conventional surgery, (c) studies that evaluated postoperative calcium and/or parathyroid hormone (PTH) levels and the incidence of hypoCa and/or hypoPT, and (d) studies in the English language.

### Exclusion criteria

The exclusion criteria were: (a) inclusion of patients who underwent hemithyroidectomy, lobectomy, or partial thyroidec-tomy; (b) use of other optical techniques, such as indocyanine green (ICG) and methylene blue for PG preservation in most surgical procedures; (c) inclusion of patients with preoperative PG disease and/or abnormal parathyroid function; (d) reviews, letters, meta-analyses, case reports, animal experiments, or basic research; and (e) duplicate publications with previous studies.

### Data extraction

The extracted information included: (a) basic data of the studies (first author, year and country, and study design); (b) characteristics of the group studied (number of patients, sex, age, diagnosis, type of NIRAF device, and the use of frozen section); (c) the incidence of T-hypoCa (calcium or corrected Ca level), T-hypoPT, P-hypoCa, and P-hypoPT (long-term data); and (d) the total number of PGs identified in situ and in resected specimens, autotransplanted PGs, and the number of inadvertently resected PGs identified by permanent pathology. Missing data were excluded from the analysis.

Since the actual total number of identified PGs was not shown in the studies by Benmiloud et al. [17], Kim et al. [25], and Van Slycke et al. [26], they were calculated from the number of identified PGs per patient multiplied by the number of patients.

For the number of inadvertently resected PGs in the studies by DiMarco et al. [27] and Lykke et al. [22], we assumed that there was one PG per patient because the number of patients was reported instead of the number of PGs. Similarly, for the number of autotransplanted PGs in the studies by Benmiloud et al. [17] and Wolf et al. [20], we assumed that there was one PG per patient because they described the number of patients who were autotransplanted with at least one PG.

### Quality assessment

The articles were classified as RCTs and non-randomized studies (NRSs) and included both prospective cohort and case-control studies. Quality assessment of the included studies was performed in duplicate by two authors (TT and JO), according to the Cochrane tool for assessing risk of bias for RCTs (version 2) [28], and the Newcastle–Ottawa Scale for NRSs [29]. Disagreements were resolved by a third investigator or by consensus among the research team. We also used funnel plots and Egger's linear regression test (applicable to >10 studies) to assess reporting bias and make the analysis more reliable.

### Analyses and statistics

The incidence of T/P-hypoCa/PT was compared between the groups using NIRAF techniques and the N-E to identify PGs. A meta-analysis was performed using EZR (Easy R; the R Foundation, Vienna, Austria) [30] to calculate the pooled odds ratio (OR) and weighted mean deviation with a random-effects model (DerSimonian–Laird). Heterogeneity was assessed using Cochran's Q test and $I^2$ statistics, and $I^2 > 50\%$ for heterogeneity was considered significant. In the analysis of the number of PGs identified, the total number of PGs including unidentified PGs per patient was calculated as four.

Meta-regression analyses were performed using the metatest function from the EZR metatest package for each outcome to identify the moderators that influenced the pooled estimates of the meta-analyses. These moderators were: 1) the incidence of T-hypoCa by N-E surgery, which may be affected by a surgeon's skill; 2) the percentage of malignant disease, which usually requires central neck dissection and autotransplantation of PGs. Moreover, the correlation analysis between the incidence of T-hypoCa by N-E surgery and the percentage of malignant disease or percentage of autotransplantation by N-E surgery was performed using Spearman's rank correlation coefficient.

All statistical tests were two-sided, and a p-value <0.05 was considered significant. The protocol for this meta-analysis is available at the UMIN (registration code: UMIN 000052850).

## Results

### Literature search and study selection

A flowchart describing the article selection process is shown in Fig 1. The literature search retrieved 387 articles (PubMed: 172, Web of Science: 170, and Cochrane Library: 45). Duplicate studies were removed; 256 unique articles were subsequently reviewed. After applying the inclusion/exclusion criteria to the abstracts, 236 articles were excluded and 20 potentially relevant full-text articles were assessed for eligibility. After reading the full-text of the articles, 14 were selected for analysis. S2 Table showed all studies identified in the literature search after the exclusion of duplicates and gives reasons for their exclusion from the meta-analysis. The study by Bergenfelz et al. [21], in which ICG angiography was used in a small proportion of cases (4/486; <1%), was included. The study by Lykke et al. [22], which included not only primary total thyroidectomies but also completion thyroidectomies, was also included.

The literature search retrieves 387 articles and 14 are selected for analysis

### Characteristics of included studies

The characteristics of the selected articles are presented in Fig 2. Of the 14 studies, 1 [21] and 3 [17,19,22] were multicenter studies from four countries and a single country, respectively. The remaining ten were single-center

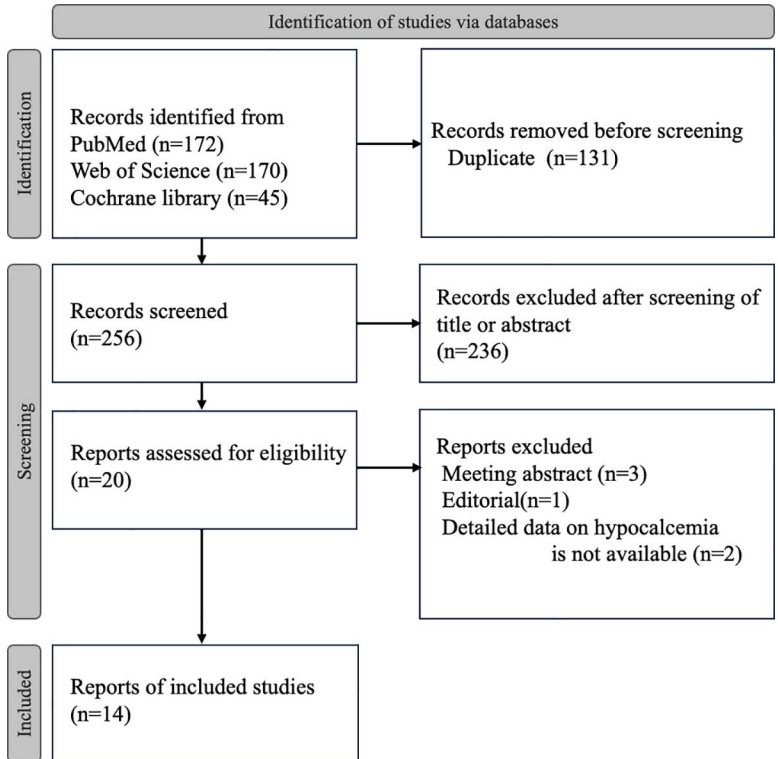

**Fig 1. Flow diagram illustrating data collection and selection of studies.**

studies. [18,20,25–27,31–35] Six were RCTs [17–22] and there were four case-control and cohort studies each. Three studies included only benign thyroid disease [20,26,35], while the remaining studies included 3.5–100% of thyroid cancer cases. In three studies frozen section was performed prior to PGs autotransplantation; however, pathology results were not described [18,22,32]. T-hypoCa was evaluated in all studies, T-hypoPT in 8, P-hypoCa in 10, and P-hypoPT in 3.

HypoCa was most frequently defined as a serum calcium level <8.0 mg/dL in six studies, while alternative definitions included a calcium level corrected for albumin <.00 mmol/L in four studies [20,21,26,27], ionized calcium level <1.09 mmol/L in one [32], <1.15 mmol/L in one [22], total calcium level <2.1 mmol/L and/or ionized calcium level <1.18 mmol/L in one [33], and an absence of a clear statement in one [34].

HypoPT was defined as a PTH level <12 pg/mL [20], <15 pg/mL [26,32,33], <20 pg/mL [19], and <1 pmol/L [27]. In addition, one study defined hypoPT as the need for treatment with a vitamin D analog or oral calcium therapy [21], and one study did not define hypoPT [34].

Of the studies evaluating P-hypoCa/PT, only one evaluated at 3 months [22]; the remaining seven evaluated at ≥6 months. Ten studies evaluated the number of identified PGs, 11 inadvertent resections, and 10 autotransplantations.

## Quality assessment and publication bias

For the six RCTs, the risk of bias assessment via the traffic light plot is shown in Fig 3. The randomization process, deviations from the intended interventions, missing outcome data, and outcome measurements were judged to have a low risk of bias in all trials. The reported results in the studies by Dip et al. [18], Papavramidis et al. [19], and Wolf et al. [20] were considered to have some concerns because some of the registry outcome scales and actual reported outcomes were

| 1st Author | Article type | Publication date | country | institution | Age (NIRAF group) | Male /Female | Diagnosis: malignant disease (%) | Instrument | Definition of hypoCa | Definition of hypoPT |
|---|---|---|---|---|---|---|---|---|---|---|
| Benmiloud [31] | Cohort study | 2018 | France | single | 49.6(mean) | NA | 18(3.5%) | Fluobeam 800 system | calcium level <8 mg/ dL | NA |
| Dip [18] | RCT | 2019 | Argentina | single | 48.7(mean) | 44/126 | 82(48.2) | Fluobeam 800 system | calcium level <8 mg/ dL | NA |
| DiMarco [27] | Cohort study | 2019 | UK | single | 48(mean) | 48/221 | 136(50.6) | Fluobeam 800 system | calcium level corrected for albumin < 2.00 mmol/l | PTH < 1 pmol/l |
| Benmiloud [17] | RCT | 2020 | France | triple in France | 52.0(median) | 49/192/1 | 38(15.8) | Fluobeam 800 system | calcium level <8 mg/ dL | NA |
| Kim YS [25] | case control | 2020 | Korea | single | 51.6(mean) | 59/241 | 134(44.7) | Fluobeam 800 system | calcium level <8 mg/ dL | NA |
| Papavramidis [19] | RCT | 2021 | Greece | double in Greece | 48.4(mean) | 47/133 | 89(49.4) | Fluobeam LX | calcium level <8 mg/ dL | PTH < 20 pg/ml |
| Van Slycke [26] | Cohort study | 2021 | Belgium | single | NA | 219/864 | 0 | Fluobeam 800 system | calcium level corrected for albumin <2.00 mmol/L | PTH <15 ng/L. |
| Kim DH [32] | case control | 2021 | Korea | single | 51.3(mean) | 99/433 | 542(100) | Original system | ion-Ca. level (<1.09 mmol/L) | PTH < 15 pg/ml |
| Wolf [20] | RCT | 2022 | Germany | single | 57 (mean) | 17/43 | 0 | Original system(Karl Storz) | absolute calcium levels lower than 2 mmol/L | PTH < 12 ng/L |
| Barbieri [33] | case control | 2022 | Italy | single | 52.1(mean) | 34/100 | 27(20.2) | Fluobeam LX | total calcium level was <2.1 mmol/Land/or ionized calcium level was <1.18mmol/L | PTH <15pg/ml |
| Lykke [22] | RCT | 2023 | Denmark | double in Denmark | 57.4(mean) | 37/110 | 88(60.0) | Elevision IR or Fluobeam 800 | ion-Ca2+ < 1.15 mmol/L | NA |
| Bergenfelz [21] | RCT | 2023 | Sweden Poland Norway Austria | quadruple in four country | 50.2(mean) | 97/389 | 116 (23.9) | Fluobeam LX | a total calcium level below 2.00 mmol/l. | The need for treatment with a vitamin D analogue or oral calcium therapy |
| Huang [34] | Cohort study | 2023 | China | single | 47.88(mean) | 26/74 | 100(100) | Jinan Microsmart Intelligence Technology Co., Ltd. | NA | NA |
| Uysal [35] | case control | 2023 | USA | single | 46.0(mean) | 31/154 | 0(0) | Fluobeam 800 system | calcium level <8 mg/ dL | NA |

**Fig 2. Characteristics of included studies.** RCT, randomized controlled trial; NIRAF, near-infrared autofluorescence; M, male; F, female; hypoCa, hypocalcemia; hypoPT, hypoparathyroidism; PTH, parathyroid hormone.

different. In the overall assessment, three studies were rated as having a low risk of bias, and the remaining three were rated as having some concerns.

For the eight NRSs, the quality of the selected literature assessed using the Newcastle–Ottawa scale is shown in Fig 4. All eight studies scored ≥7/9, indicating that most of the included literature exhibited good quality.

No asymmetry was observed in the funnel plots for T/P-hypoCa/PT (Fig 5). The p-values derived from the Egger's test were 0.8363 and 0.279 for T-hypoCa and P-hypoCa, respectively.

Asymmetry is not observed in the funnel plots of T/P-hypoCa/PT. P value of Egger's test for T-hypoCa and P-hypoCa is 0.8363 and 0.279, respectively.

## Meta-analysis

For T-hypoCa, 4,281 patients were included in this analysis (1,391 with NIRAF use; 2,890 with N-E) from six RCTs and eight NRSs. The pooled OR of all studies was 0.56, demonstrating statistical significance (95% confidence interval [CI]: 0.43–0.72, $p < 0.001$). Significant heterogeneity was not found because the $I^2$ score = 20%, and Cochrane's Q test showed p = 0.24 (Fig 6).

The experimental arm underwent parathyroid near-infrared autofluorescence, whereas the control arm underwent conventional naked-eye surgery. Diamonds correspond to the pooled odds ratio with a relative confidence interval of 95%.

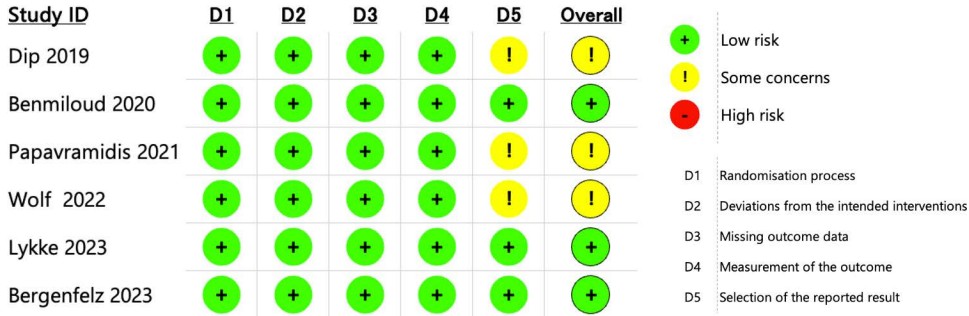

**Fig 3. Risk-of-bias assessment for included studies.** Green, low risk of bias; yellow, moderate risk of bias; red, high risk of bias.

| Newcastle-ottawa scale for nonRCT | | | | | | | | | |
|---|---|---|---|---|---|---|---|---|---|
| **Cohort studies** | | | | | | | | | |
| First author/year | Representativeness of the exposed cohort | Selection of control | Ascertainment of exposure | Outcome of Interest not present at start of the study | Comparability of controls | Assessment of outcome | Sufficient follow-up duration | Adequacy of follow-up | Overall bias |
| Benmiloud/2018 | * | / | * | * | ** | * | * | * | 8/9 |
| DiMarco/2019 | / | * | * | * | * | * | * | * | 7/9 |
| Van Slycke/2021 | * | * | * | * | ** | * | * | * | 9/9 |
| Huang/2023 | * | * | * | * | * | * | * | * | 8/9 |
| | | | | | | | | | |
| **Case control studies** | | | | | | | | | |
| First author/year | Adequacy of case definition | Representativeness of the cases | Selection of controls | Definition of controls | Comparability of cases and controls | Ascertainment of exposure | Same method of ascertainment for cases and controls | Non-Response rate | Overall bias |
| Kim YS/2020 | * | * | * | * | ** | * | * | * | 9/9 |
| Kim DH/2021 | * | * | * | * | * | * | * | * | 8/9 |
| Barbieri/2022 | * | * | / | * | * | * | * | * | 7/9 |
| Uysal/2023 | * | * | * | * | ** | * | * | * | 9/9 |

**Fig 4. Newcastle–Ottawa scale for nonrandomized studies.** RCT: randomized control trail.

For T-hypoPT, 2725 patients were included in this analysis (838 with NIRAF use; 1887 with N-E use) from three RCTs and five NRSs. The pooled OR of all studies was 0.56, demonstrating statistical significance (95% CI: 0.40–0.79, p < 0.001). Significant heterogeneity was found with an $I^2$ score = 51% and Cochrane's Q test of p = 0.04 (Fig 6).

For P-hypoCa, 3,538 patients were included in this analysis (1,018 with NIRAF; 2,520 with N-E) from three RCTs and eight NRSs. The pooled OR was 0.61, demonstrating no statistical significance (95% CI: 0.33–1.13, p = 0.1156). Heterogeneity was not found because the $I^2$ score = 0%, and Cochrane's Q test showed p = 0.99 (Fig 7).

The experimental arm underwent parathyroid near-infrared autofluorescence, whereas the control arm underwent conventional naked-eye surgery. Diamonds correspond to the pooled odds ratio with a relative confidence interval of 95%.

For P-hypoPT, 1,161 patients were included in this analysis (573 with NIRAF; 588 with N-E) from one RCT and two NRSs. The pooled OR of all studies was 0.80, demonstrating no statistical significance (95% CI: 0.44–1.43, p = 0.4461). Heterogeneity was not found because the $I^2$ score = 0%, and Cochrane's Q test showed p = 0.75 (Fig 7).

For the number of PGs identified in situ and in resected specimens, the pooled OR from 10 studies was 1.45, demonstrating statistical significance (95% CI: 1.04–2.04, p = 0.030, Fig 8). For inadvertent resection of PGs identified by the

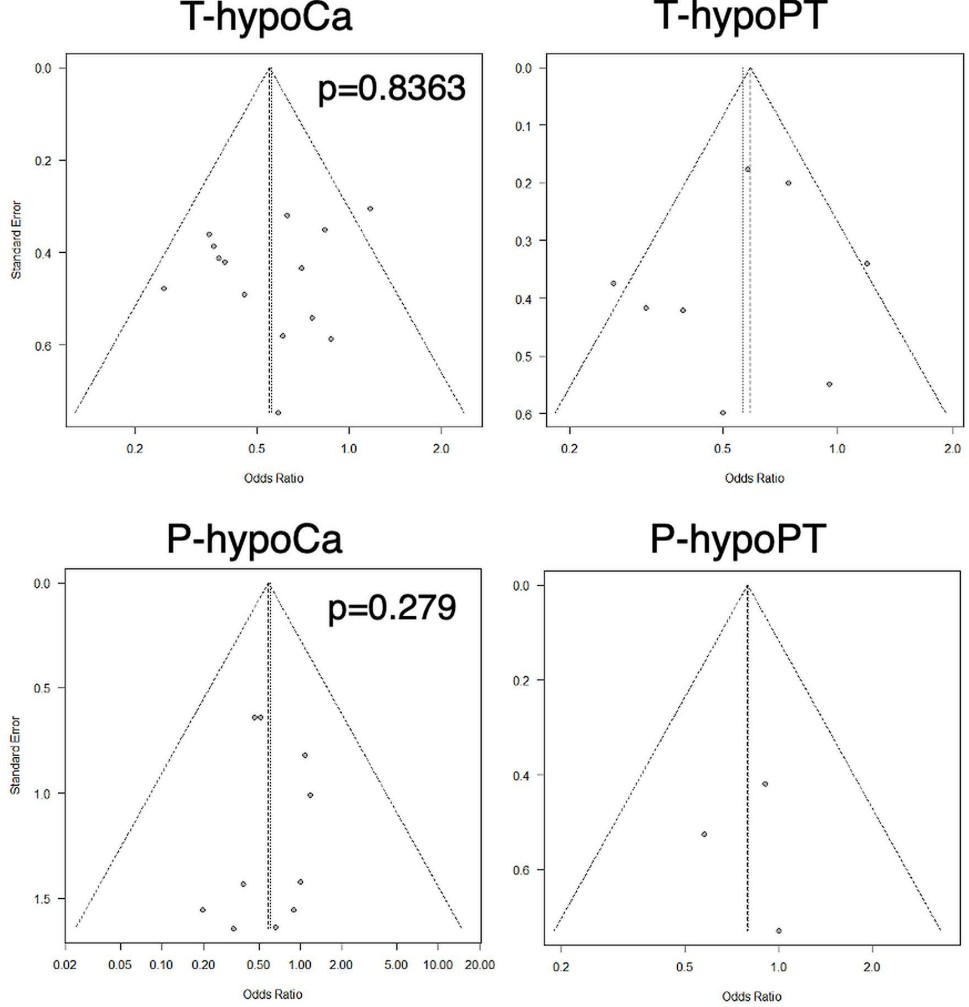

**Fig 5. Funnel plots of Temporary/Permanent- hypocalcemia/parathyroidism (T/P-hypoCa/PT).**

permanent pathology, the pooled OR from 11 studies was 0.45, demonstrating statistical significance (95% CI: 0.26–0.78, p = 0.004, Fig 8). For autotransplantation of PGs, the pooled OR from 10 studies was 0.87, demonstrating no statistical significance (95% CI: 0.45–1.66, p = 0.6653, Fig 8).

The experimental arm corresponds to parathyroid near-infrared autofluorescence use, while the control arm to conventional naked-eye surgery.

**Meta-regression analysis and correlation analysis**

Meta-regression analyses were conducted on T/P-hypoCa, T/P-hypoPT, the number of PGs identified in situ and in resected specimens, inadvertent resection, and autotransplantation of PGs concerning (a) the incidence of T-hypoCa using conventional searching methods relying on N-E; (b) the percentage of malignant disease.

## Temporary Hypocalcemia

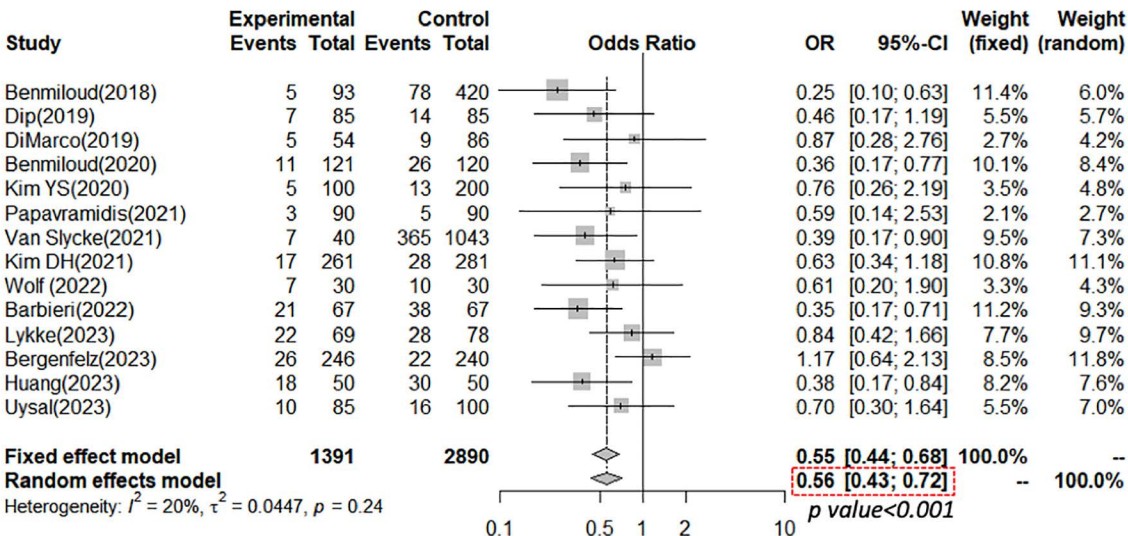

## Temporary Hypoparathyroidism

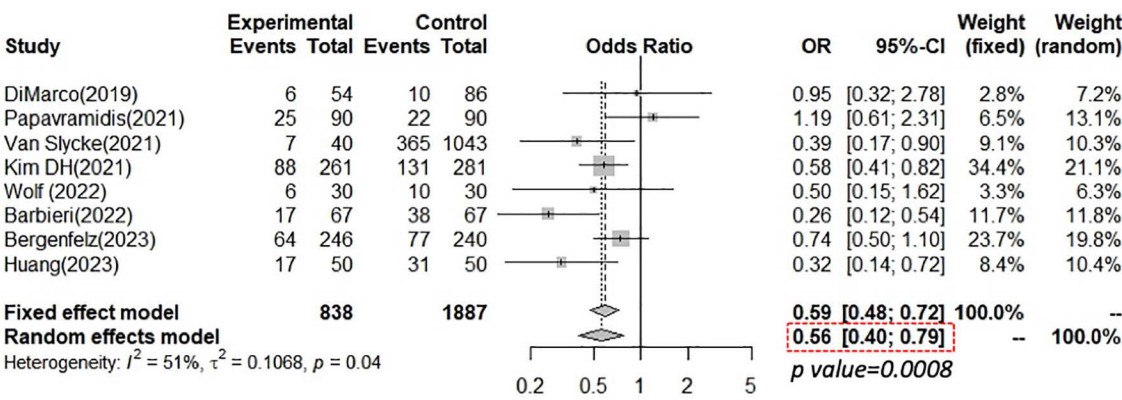

**Fig 6. Forest plot of temporary hypocalcemia and temporary hypoparathyroidism.**

Significant negative correlations were found between the incidence of T-hypoCa by N-E surgery and the ORs for T-hypoCa, T-hypoPT, and PGs autotransplantation (p = 0.0385, 0.00192, and 0.0225, respectively, Fig 9A,9B and 9E). In contrast, other moderators showed no significant associations, including those between the percentage of malignant disease and the ORs for T-hypoCa and T-hypoPT (p = 0.61and p = 0.90, respectively, Fig 9C and 9D).

Correlation analysis revealed no correlation between the percentage of malignant diseases and incidence of T-hypoCa by N-E surgery, whereas moderate correlation was found between the incidence of T-hypoCa and percentage of autotransplantation in N-E surgeries (r = 0.624, p = 0.0602) (Fig 10B).

The correlation analysis between the incidence of temporary hypocalcemia (T-hypoCa) by the naked-eye (N-E) surgery and percentage of malignant disease (A), and percentage of PG autotransplantation and incidence of T-hypoCa by N-E surgery (B).

We further performed stratified analysis by the ORs of Group A (incidence of T-hypoCa with N-E >15%) and Group B (<15%) were 0.45 (95% CI: 0.35–0.60) and 0.84 (95% CI: 0.58–1.12), respectively, showing a significant difference between the two groups (p = 0.0091) (Fig 11).

The incidence was >15% in Group A and <15% in Group B

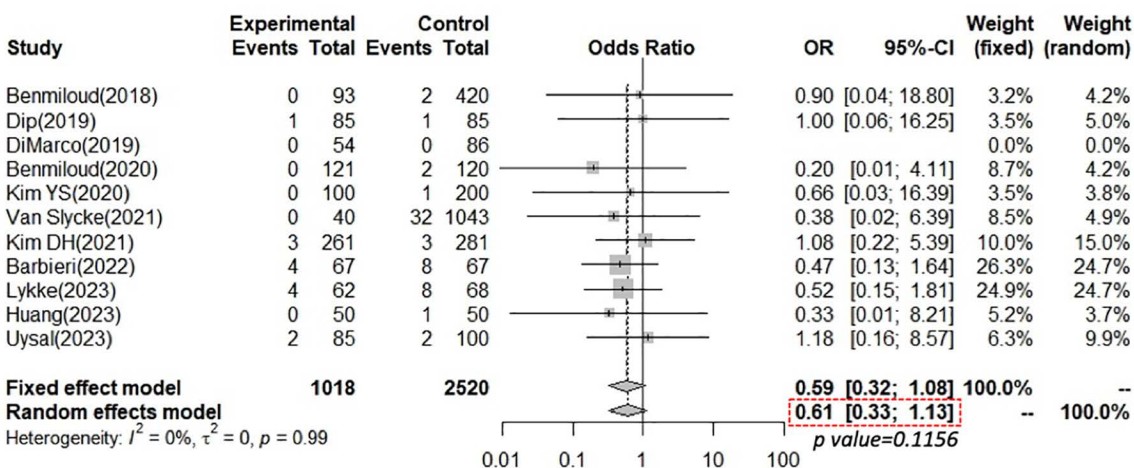

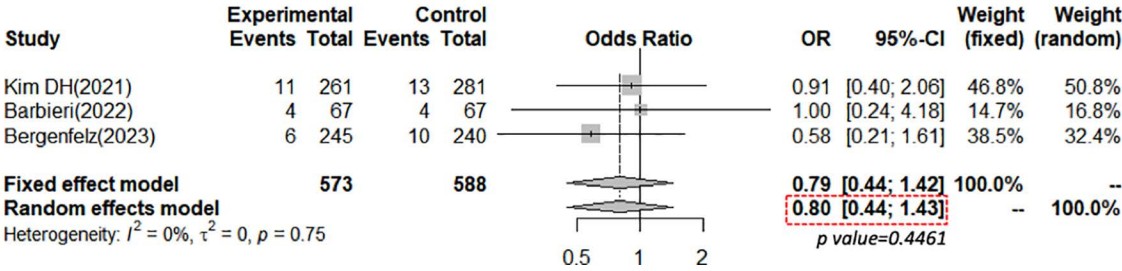

**Fig 7. Forest plot of permanent hypocalcemia and permanent hypoparathyroidism.**

# Number of parathyroid glands identified
## in situ and in resected specimens

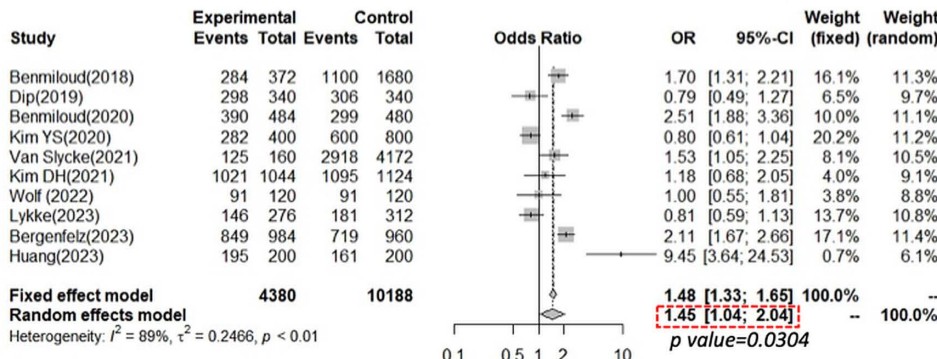

# Inadvertent resection of parathyroid glands

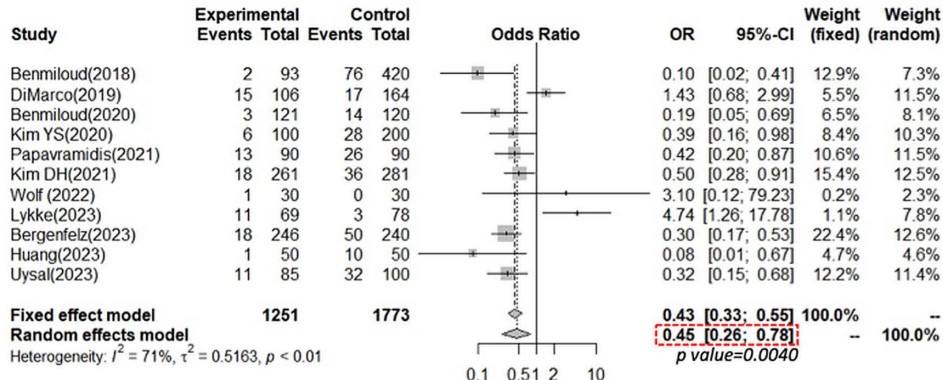

# Autotransplantation of parathyroid glands

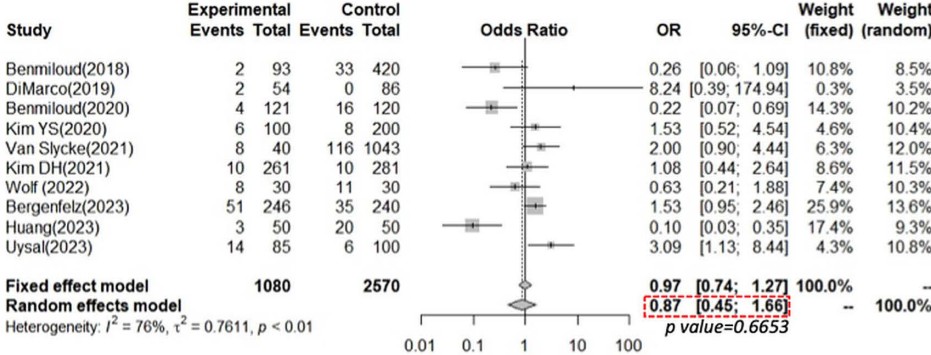

**Fig 8. Forest plot of meta-analysis for the number of parathyroid glands identified, inadvertent resection of parathyroid glands, and auto-transplantation of parathyroid glands.**

## Discussion

### Quality assessment

Traffic light plot for RCTs (Fig 3) and the Newcastle–Ottawa scale for NRSs (Fig 4) revealed that this meta-analysis did not suffer from serious bias and included only high-quality studies. Funnel plots and the Egger's test demonstrated no publication bias for T/P-hypoCa/PT. These data support the results of our meta-analysis.

### Temporary-hypoCa/PT

Our study, comprising 14 studies and 4,281 surgical procedures, demonstrated that the NIRAF use significantly reduced the incidence of T-hypoCa/PT: the pooled ORs for T-hypoCa/PT were 0.56 (95% CI: 0.43–0.72, p < 0.001) and 0.56 (95% CI: 0.40–0.79, p < 0.001), respectively. Given that the low incidence of T-hypoPT directly indicates the preservation of parathyroid function, NIRAF seems to contribute to the in situ preservation of PGs, resulting in the prevention of T-hypoCa.

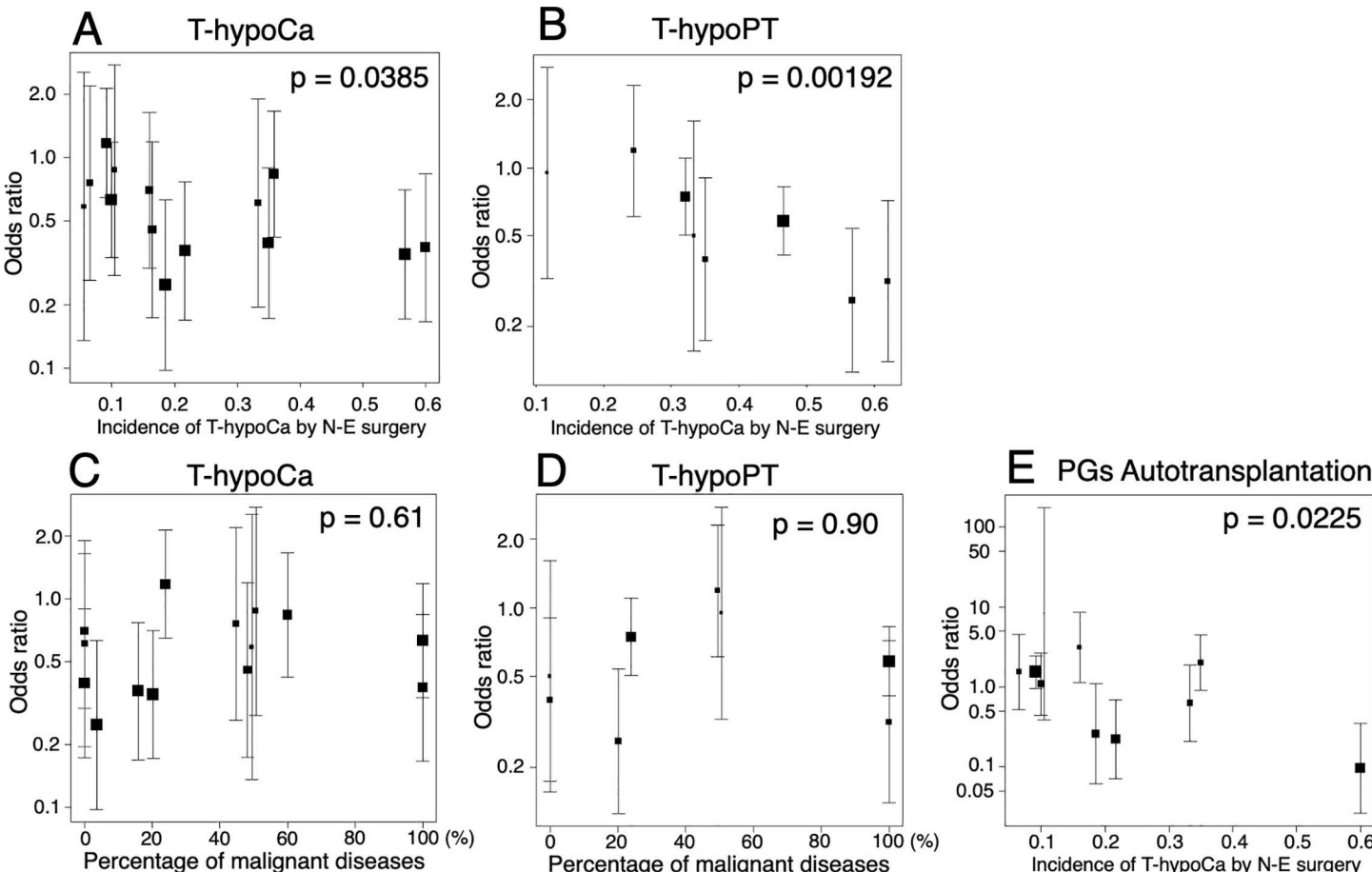

**Fig 9. Meta-regression analysis.** Meta-regression analysis of temporary hypocalcemia (T-hypoCa) (A)/ temporary hypoparathyroidism (T-hypoPT) (B) with the incidence of T-hypoCa by the naked-eye (N-E) surgery, T-hypoCa (C)/ T-hypoPT (D) with the percentage of malignant diseases, and autotransplantation of parathyroid glands (PGs) with the incidence of T-hypoCa by N-E surgery (E).

The benefits of NIRAF may depend on 1) surgeons' skills: relatively less beneficial for experienced and skillful surgeons who can identify PGs to preserve parathyroid function even without NIRAF use and 2) pathologies: relatively more beneficial in malignant cases to identify PGs for autotransplantation in excised specimens following central neck lymph node

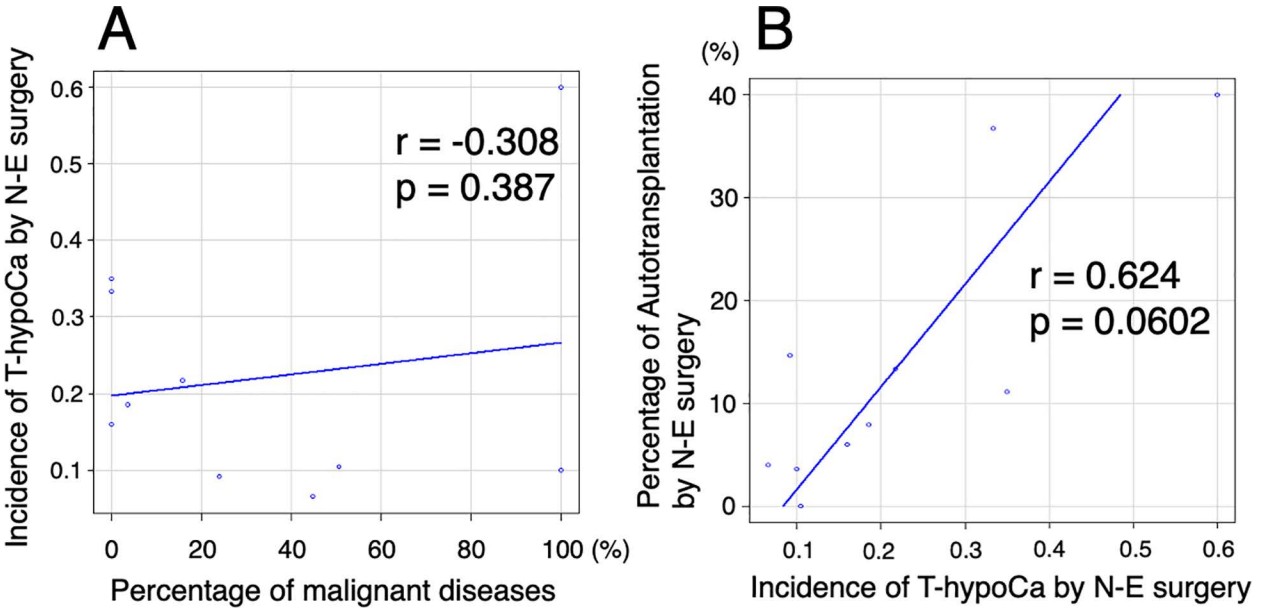

**Fig 10. The correlation analysis.**

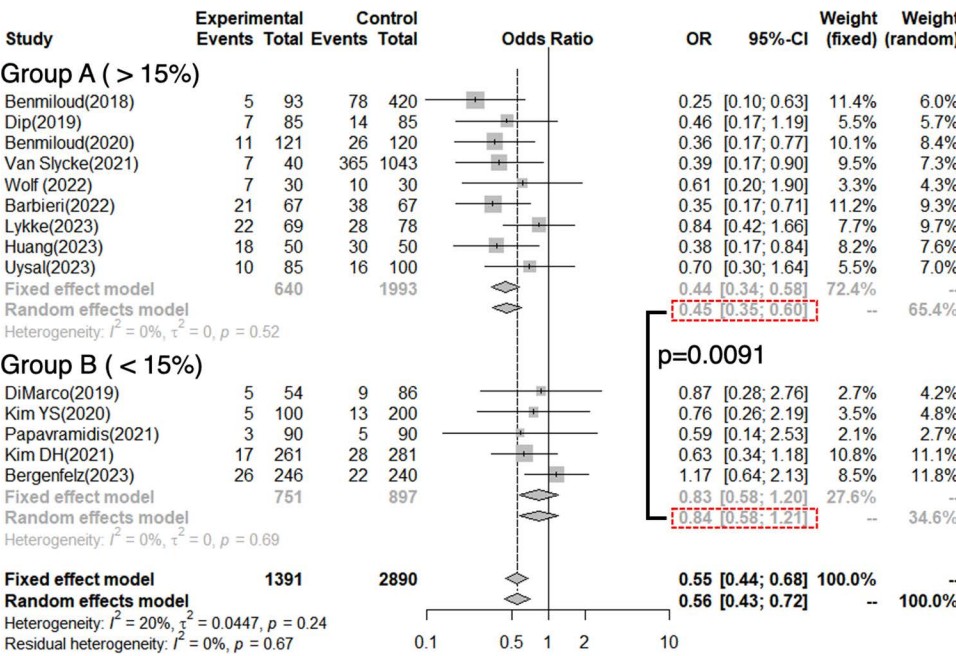

**Fig 11. The incidence of temporary hypocalcemia (T-hypoCa) with parathyroid near-infrared autofluorescence use is analyzed by the incidence of T-hypoCa by the naked-eye (N-E) surgery.**

dissection. Therefore, regarding 1), we tested the association between the incidence of T-hypoCa by N-E surgery, which may be affected by surgeons' skill and ORs of T-hypoCa/hypoPT. Regarding 2), association between the percentage of malignant diseases and ORs of T-hypoCa/hypoPT was also examined. Meta-regression analyses revealed that the higher the incidence of T-hypoCa by N-E surgery, the lower the ORs of T-hypoCa/hypoPT with NIRAF use (Fig 9A and 9B). Thus, NIRAF may be particularly beneficial for less skilled surgeons with a high incidence of T-hypoCa by N-E surgery in preventing T-hypoCa/hypoPT. In contrast, there was no association between the percentage of malignant disease and ORs of T-hypoCa/hypoPT (Fig 9C and 9D). Moreover, percentage of malignant diseases did not correlate with incidence of T-hypoCa by N-E surgery (Fig 10A). Further stratified analysis also showed that NIRAF use resulted in a significant improvement in T-hypoCa for surgeons with an incidence of ≥15% of T-hypoCa by N-E (Fig 11). These results suggest that the use of NIRAF may especially benefit less skilled surgeons to prevent T-hypoCa/hypoPT and pathology and its related surgical procedures do not largely affect the usefulness of NIRAF.

## Permanent HypoCa

Although no significant differences were noted in P-hypoCa between the NIRAF and the N-E groups, there was a tendency of NIRAF use in reducing the incidence of P-hypoCa. The incidence of P-hypoCa in both the NIRAF and the N-E groups was as low as 1.4% (14/1018) and 2.4% (60/2520), respectively. Therefore, we considered that the non-significant results may be due to the insufficient sample size. Assuming a P-hypoPT rate of 2.4% in the N-E group and 1.4% in the NIRAF group, a power of 0.8, and a significance of 0.05, a total of 6250 patients (3,125 per group) would be needed to generate a statistically meaningful result. In addition, since recent epidemiological studies using national registries, insurance databases, or multicenter studies have shown a higher incidence rate of P-hypoPT, ranging from 11.2–15.0% [5–9], surgeons who participated in the present study were considered relatively experienced and highly skilled in preserving parathyroid function even without NIRAF use. The role of NIRAF in P-hypoCa/PT should be the subject of future studies using national registries or insurance databases that would include data from surgeons with varying skills.

## Impact of NIRAF on PG autotransplantation

Autotransplantation of PGs may be the second-best method following in situ preservation to prevent P-hypoCa/hypoPT. As shown in Fig 8, the pooled data demonstrated a significant increase in the number of PGs identified in situ or in the resected specimens as well as a significant decrease in inadvertent resections of PGs by the use of NIRAF. However, the OR of autotransplantation of PGs was not significantly higher by the use of NIRAF (Fig 8), probably because NIRAF was as useful for in situ preservation of PGs as for detecting PGs eligible for autotransplantation. In N-E surgeries, incidence of T-hypoCa showed moderate positive correlation with the percentage of autotransplantation (Fig 10B). Nonetheless, meta-regression analysis showed that the ORs for T-hypoCa/T-hypoPT and the PGs autotransplantion with NIRAF use were negatively associated with the incidence of T-hypoCa by N-E surgery (Fig 9A,B,E). This implies that the use of NIRAF can be useful to increase the number of PGs that can be preserved in situ and decrease the number of autotransplanted PGs in the high incidence group of T-hypoCa by N-E surgery. At the same time, NIRAF may be useful in the low incidence group of T-hypoCa by N-E surgery because it aids in detecting PGs within excised specimens, thereby increasing the number of autotransplanted PGs (Fig 9E). Therefore, the OR for autotransplanted PGs was non-significant when surgeons with several incidence of T-hypoCa by N-E surgery were mixed (Fig 8).

## Limitations

This study had some limitations. First, P-hypoPT was assessed in only three studies. Therefore, further investigations are necessary to collect and consolidate information on parathyroid function, particularly focusing on PTH levels not only immediately after surgery but also in the long term. Second, the incidence of T-hypoCa by N-E surgery is thought to

largely depend on the surgeon's skill in PG in situ preservation, but other factors that are currently unconsidered, such as the extent of lymph node dissection and strategy of autotransplantation, may also be relevant. Third, the results of the frozen section analysis might have influenced the decision to perform autotransplantation. However, these results were not described in all studies. Finally, the absence of a standardized measurement method and cutoff values for calcium and PTH levels may have influenced our results.

## Conclusions

Our results indicated a significant difference in the incidence of T-hypoCa and T-hypoPT between the NIRAF and N-E groups after total thyroidectomy. Meta-regression analysis revealed that NIRAF benefits surgeons with the high incidence of T-hypoCa by N-E surgery (≥15%) by reducing T-hypoCa/PT and skillful surgeons by increasing the number of autotransplanted PGs. From these results, it is evident that the use of NIRAF in total thyroidectomy provides significant benefits for surgeons with different levels of experience, such as novices and experts.

## Supporting information

**S1 Table. PRISMA_2020 checklist.**
(XLSX)

**S2 Table. Data extracted from included studies and used for all analyses.**
(TIFF)

**S3 Table. The concise dataset of this study.**
(DOCX)

## Acknowledgments

We would like to thank Editage (www.editage.jp) for English language editing.

## Author contributions

**Conceptualization:** Takeshi Takahashi.

**Data curation:** Takeshi Takahashi, Shusuke Ohshima.

**Formal analysis:** Takeshi Takahashi, Ryohei Oya.

**Investigation:** Takeshi Takahashi.

**Supervision:** Ryohei Oya, Hidenori Inohara, Arata Horii.

**Writing – original draft:** Takeshi Takahashi, Shalyn J. D Sa, Jo Omata.

**Writing – review & editing:** Takeshi Takahashi, Shalyn J. D Sa, Shusuke Ohshima, Jo Omata, Yusuke Yokoyama, Ryusuke Shodo, Yushi Ueki, Yukinori Takenaka, Hidenori Inohara, Arata Horii.

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
