## [Decision Letter · Decision Letter 0]

14 Nov 2024

PONE-D-24-42113Parathyroid near-infrared autofluorescence differently benefits depending on the surgeon's skill for preventing from hypoparathyroidism after total thyroidectomy: A systematic review and meta-analysisPLOS ONE

Dear Dr. Takahashi,

Thank you for submitting your manuscript to PLOS ONE. After careful consideration, we feel that it has merit but does not fully meet PLOS ONE’s publication criteria as it currently stands. Therefore, we invite you to submit a revised version of the manuscript that addresses the points raised during the review process.

Dear Authors,

Thank you for submitting your manuscript for consideration. Your study investigating the impact of parathyroid near-infrared autofluorescence on hypoparathyroidism after total thyroidectomy is well-conducted, and your analysis highlights important findings in this area. However, to further strengthen the manuscript and address key concerns, the following revisions are necessary:

**Frozen Section Analysis:**The impact of using frozen sections on hypoparathyroidism outcomes requires further clarification. Please address how this variable may influence your results and whether it was explicitly accounted for in the studies analyzed. If applicable, include a sensitivity analysis or a discussion of its limitations.**Surgeon Skill and Procedural Variability:**It is challenging to attribute the incidence of transient hypocalcemia solely to surgical skill using the current framework. Consider revising the discussion to distinguish between surgical skill and variability in surgical procedures (e.g., lymph node resection techniques or parathyroid autotransplantation). Additional analysis demonstrating the relationship between surgical skill and outcomes would strengthen this aspect.**Modeling Autotransplantation and Inadvertent Resection:**Since assumptions about the rates of parathyroid autotransplantation and inadvertent resection influence your results, please clarify their impact using sensitivity analysis or modeling. Explicitly state the assumptions and their implications on the overall conclusions.**Highlighting Efficacy of Near-Infrared Autofluorescence:**Your manuscript effectively discusses the benefits of parathyroid near-infrared autofluorescence. Ensure these findings are emphasized in the results and discussion sections. A summary table or figure may further enhance the clarity and impact of your findings.

By addressing these points, the manuscript will provide a more comprehensive and robust contribution to the field. We look forward to your revised submission.

We look forward to receiving your revised manuscript.

Kind regards,

Mohamed Khaled Ibrahim, MD

Guest Editor

PLOS ONE

Journal Requirements:

2. In this instance it seems there may be acceptable restrictions in place that prevent the public sharing of your minimal data. However, in line with our goal of ensuring long-term data availability to all interested researchers, PLOS’ Data Policy states that authors cannot be the sole named individuals responsible for ensuring data access (http://journals.plos.org/plosone/s/data-availability#loc-acceptable-data-sharing-methods). Data requests to a non-author institutional point of contact, such as a data access or ethics committee, helps guarantee long term stability and availability of data. Providing interested researchers with a durable point of contact ensures data will be accessible even if an author changes email addresses, institutions, or becomes unavailable to answer requests. Before we proceed with your manuscript, please also provide non-author contact information (phone/email/hyperlink) for a data access committee, ethics committee, or other institutional body to which data requests may be sent. If no institutional body is available to respond to requests for your minimal data, please consider if there any institutional representatives who did not collaborate in the study, and are not listed as authors on the manuscript, who would be able to hold the data and respond to external requests for data access? If so, please provide their contact information (i.e., email address). Please also provide details on how you will ensure persistent or long-term data storage and availability.

3. Please include a new copy of Tables1 and 2 in your manuscript; the current table is difficult to read. Please follow the link for more information: https://blogs.plos.org/plos/2019/06/looking-good-tips-for-creating-your-plos-figures-graphics/ "

4. As required by our policy on Data Availability, please ensure your manuscript or supplementary information includes the following:

Reviewers' comments:

Reviewer's Responses to Questions

**Comments to the Author**

1. Is the manuscript technically sound, and do the data support the conclusions?

Reviewer #1: Yes

Reviewer #2: Yes

2. Has the statistical analysis been performed appropriately and rigorously? 

Reviewer #1: Yes

Reviewer #2: Yes

3. Have the authors made all data underlying the findings in their manuscript fully available?

Reviewer #1: Yes

Reviewer #2: Yes

4. Is the manuscript presented in an intelligible fashion and written in standard English?

Reviewer #1: Yes

Reviewer #2: Yes

5. Review Comments to the Author

Reviewer #1: Thank you for allowing me to conduct the following review.

The article attempts to summarize information about the use of near-infrared autofluorescence (NIRAF) of parathyroid glands during thyroidectomy from selected literature on this subject. The conclusions supported by a thorough statistical analysis and a discussion of the limitations of the method used will allow the potential reader to learn more about the application and effectiveness of the described method for detecting parathyroid glands. I believe that the article in its current form qualifies for publication.

Reviewer

Reviewer #2: Thank you for giving me the opprotunity to review this manuscript. This study investigated the impact of the parathyroid near-infrared autofluorescence on the hypoparathyroidism after total thyroidectomy using systematic review and meta analysis. The analysis of this study was conducted well. However, the impact of the frozen section of the parathyorid glands might influecnce the results. It is uncertain to define the surgeons' skill using incidence of T-hypo Ca relying on the N-E, because in the lymph node resection, some surgeons resect all of the parathyroid glands with lymp nodes and autografted. This might depend on the surgical procedures, not the surgical skills. The authors should demonstrate the results that surgeons' skill can cause T-hypo Ca relying on the N-E. The authors should clarify the influence of assumption of numbers of autotransplanted and inadvertently resected on the results using some models because they are only assumptions. However, this manuscript well mentioned the efficacy of parathyroid near-infrared autofluorescence.

6. PLOS authors have the option to publish the peer review history of their article (what does this mean? ). If published, this will include your full peer review and any attached files.

**Do you want your identity to be public for this peer review?** For information about this choice, including consent withdrawal, please see our Privacy Policy .

Reviewer #1: No

Reviewer #2: **Yes: ** Takahisa Hiramitsu

---

## [Author Response · Author response to Decision Letter 1]

9 Jan 2025

Thank you very much for your insightful comments. We have provided point-by-point responses below.

Frozen Section Analysis:

The impact of using frozen sections on hypoparathyroidism outcomes requires further clarification. Please address how this variable may influence your results and whether it was explicitly accounted for in the studies analyzed. If applicable, include a sensitivity analysis or a discussion of its limitations.

Response: Thank you for pointing out this important issue. As you have mentioned, we think it would affect the outcome, especially where a frozen section is performed before autotransplantation. Frozen sections were performed in three studies; however, the results were not described. We added the following sentence to the Results and Limitations sections.

In three studies frozen section was performed prior to PGs autotransplantation; however, pathology results were not described [18, 22, 32].

Third, the results of the frozen section analysis might have influenced the decision to perform autotransplantation. However, these results were not described in all studies.

Surgeon Skill and Procedural Variability:

It is challenging to attribute the incidence of transient hypocalcemia solely to surgical skill using the current framework. Consider revising the discussion to distinguish between surgical skill and variability in surgical procedures (e.g., lymph node resection techniques or parathyroid autotransplantation). Additional analysis demonstrating the relationship between surgical skill and outcomes would strengthen this aspect.

Response: As the reviewer has pointed out, we recognized that other factors than surgical skills, such as lymph node dissection and autotransplantation, would affect the transient hypocalcemia (T-hypoCa). Unfortunately, owing to the lack of data, it was not possible to estimate the effect of surgical procedures on T-hypoCa; therefore, we alternatively investigated the association of percentage of malignant diseases, which usually require central neck dissection and autotransplantation of PGs, with T-hypoCa. For this purpose, meta-regression analysis and correlation analysis were newly performed for malignant disease as possible factors which would influence the incidence of T-hypoCa. As a result, however, there was no association between the percentage of malignant disease and either the incidence of T-hypoCa by N-E surgery or the odds ratio (ORs) of T-HypoCa/hypoPT (Fig 8A, 7C, D). These results suggest that the incidence of T-hypoCa by the N-E surgery may be mainly related to the surgeon's skill to preserve the blood supply to the PG. We added following text in the revised manuscript.

In Methods section

These moderators were: 1) the incidence of T-hypoCa by N-E surgery, which may be affected by a surgeon’s skill; 2) the percentage of malignant disease, which usually requires central neck dissection and autotransplantation of PGs.

Moreover, the correlation analysis between the incidence of T-hypoCa by N-E surgery and the percentage of malignant disease or percentage of autotransplantation by N-E surgery was performed using Spearman’s rank correlation coefficient.

In the Results section

Significant negative correlations were found between the incidence of T-hypoCa by N-E surgery and the ORs for T-hypoCa, T-hypoPT, and PGs autotransplantation (p = 0.0385, 0.00192, and 0.0225, respectively, Fig 7A, B, E). In contrast, other moderators showed no significant associations, including those between the percentage of malignant disease and the ORs for T-hypoCa and T-hypoPT (p = 0.61and p = 0.90,　respectively, Fig 7 C, D).

Correlation analysis revealed no correlation between the percentage of malignant diseases and incidence of T-hypoCa by N-E surgery (Fig 8A), whereas moderate correlation was found between the incidence of T-hypoCa by the N-E surgery and percentage of autotransplantation by N-E surgery (Fig 8B, r = 0.624, p = 0.0602)

Figure 7

Figure 8

In Discussion

The benefits of NIRAF may depend on 1) surgeons’ skills: relatively less beneficial for experienced and skillful surgeons who can identify PGs to preserve parathyroid function even without NIRAF use and 2) pathologies: relatively more beneficial in malignant cases to identify PGs for autotransplantation in excised specimens following central neck lymph node dissection. Therefore, regarding 1), we tested the association between the incidence of T-hypoCa by N-E surgery, which may be affected by surgeons’ skill and ORs of T-hypoCa/hypoPT. Regarding 2), association between the percentage of malignant diseases and ORs of T-hypoCa/hypoPT was also examined. Meta-regression analyses revealed that the higher the incidence of T-hypoCa by N-E surgery, the lower the ORs of T-hypoCa/hypoPT with NIRAF use (Fig 7A, B). Thus, NIRAF may be particularly beneficial for less skilled surgeons with a high incidence of T-hypoCa by N-E surgery in preventing T-hypoCa/hypoPT. In contrast, there was no association between the percentage of malignant disease and ORs of T-hypoCa/hypoPT (Fig 7C, D). Moreover, percentage of malignant diseases did not correlate with incidence of T-hypoCa by N-E surgery (Fig 8A). Further stratified analysis also showed that NIRAF use resulted in a significant improvement in T-hypoCa for surgeons with an incidence of ≥15% of T-hypoCa by N-E surgery (Fig 9). These results suggest that the use of NIRAF may especially benefit less skilled surgeons to prevent T-hypoCa/hypoPT and pathology and its related surgical procedures do not largely affect the usefulness of NIRAF.

Modeling Autotransplantation and Inadvertent Resection:

Since assumptions about the rates of parathyroid autotransplantation and inadvertent resection influence your results, please clarify their impact using sensitivity analysis or modeling. Explicitly state the assumptions and their implications on the overall conclusions.

Response: As shown in Fig 6, although the number of PGs identified either in situ or in resected specimens increased and the inadvertent resection of PGs decreased by the use of NIRAF, the number of autotransplantation of PGs was not affected by the use of NIRAF. We assumed that this is perhaps due to the increase in the number of PGs that can be preserved in situ by NIRAF. For this purpose, correlation analysis was newly performed between the incidence of T-hypoCa and percentage of autotransplanations in N-E surgeries.

Correlation analysis revealed a moderate correlation between the percentage of autotransplantation of PGs and incidence of T-hypoCa by N-E surgery (Fig 8B). Therefore, in N-E surgeries, more autotransplantations of PGs were performed in surgeries with T-hypoCa. Nonetheless, the higher the incidence of T-hypoCa by N-E surgery, the lower the ORs of T-hypoCa/hypoPT and autotransplantations of PGs (Fig 7A, B, E), which means more PGs were preserved in situ by the use of NIRAF.

We added the following text in each section.

In Methods section,

These moderators were: 1) the incidence of T-hypoCa by N-E surgery, which may be affected by a surgeon’s skill; 2) the percentage of malignant disease, which usually requires central neck dissection and autotransplantation of PGs.

Moreover, the correlation analysis between the incidence of T-hypoCa by N-E surgery and the percentage of malignant disease or percentage of autotransplantation by N-E surgery was performed using Spearman’s rank correlation coefficient.

In Results section,

Correlation analysis revealed no correlation between the percentage of malignant diseases and incidence of T-hypoCa by N-E surgery, whereas moderate correlation was found between the incidence of T-hypoCa and percentage of autotransplantation in N-E surgeries (r = 0.624, p = 0.0602) (Fig 8B).

In Discussion section,

Autotransplantation of PGs may be the second-best method following in situ preservation to prevent P-hypoCa/hypoPT. As shown in Fig 6, the pooled data demonstrated a significant increase in the number of PGs identified in situ or in the resected specimens and a significant decrease in inadvertent resections of PGs by the use of NIRAF. However, the OR of autotransplantation of PGs was not significantly high by the use of NIRAF (Fig 6), probably because NIRAF was as useful for in situ preservation of PGs as for detecting PGs eligible for autotransplantation. In N-E surgeries, incidence of T-hypoCa showed moderate positive correlation with the percentage of autotransplantation (Fig 8B). Nonetheless, meta-regression analysis showed that the ORs for T-hypoCa/T-hypoPT and the PGs autotransplantion with NIRAF use were negatively associated with the incidence of T-hypoCa by N-E surgery (Fig 7A, B, E). This implies that the use of NIRAF can be useful to increase the number of PGs that can be preserved in situ and decrease the number of autotransplanted PGs in the high incidence group of T-hypoCa by N-E surgery. At the same time, NIRAF may be useful in the low incidence group of T-hypoCa by N-E surgery because it aids in detecting PGs within excised specimens, thereby increasing the number of autotransplanted PGs (Fig 7E). Therefore, the OR for autotransplanted PGs was non-significant when surgeons with several incidence of T-hypoCa by N-E surgery were mixed (Fig 6).

Highlighting Efficacy of Near-Infrared Autofluorescence:

Your manuscript effectively discusses the benefits of parathyroid near-infrared autofluorescence. Ensure these findings are emphasized in the results and discussion sections. A summary table or figure may further enhance the clarity and impact of your findings.

Response: Thank you very much for your insightful suggestions.

As you have pointed out, we thought that our results should emphasize that NIRAF is useful for many surgeons. We added the following sentence to the Conclusions section.

From these results, it is evident that the use of NIRAF in total thyroidectomy provides significant benefits for surgeons with different levels of experience, such as novices and experts.

Journal Requirements:

Response: Thank you very much for sharing the details. We have revised the manuscript according to The PLOS ONE style.

2. In this instance it seems there may be acceptable restrictions in place that prevent the public sharing of your minimal data. However, in line with our goal of ensuring long-term data availability to all interested researchers, PLOS’ Data Policy states that authors cannot be the sole named individuals responsible for ensuring data access (http://journals.plos.org/plosone/s/data-availability#loc-acceptable-data-sharing-methods). Data requests to a non-author institutional point of contact, such as a data access or ethics committee, helps guarantee long term stability and availability of data. Providing interested researchers with a durable point of contact ensures data will be accessible even if an author changes email addresses, institutions, or becomes unavailable to answer requests. Before we proceed with your manuscript, please also provide non-author contact information (phone/email/hyperlink) for a data access committee, ethics committee, or other institutional body to which data requests may be sent. If no institutional body is available to respond to requests for your minimal data, please consider if there any institutional representatives who did not collaborate in the study, and are not listed as authors on the manuscript, who would be able to hold the data and respond to external requests for data access? If so, please provide their contact information (i.e., email address). Please also provide details on how you will ensure persistent or long-term data storage and availability.

Response: The general secretary of our department will keep the data independently of the author and respond to external requests.

Email: okayoko@med.niigata-u.ac.jp

3. Please include a new copy of Tables1 and 2 in your manuscript; the current table is difficult to read. Please follow the link for more information: "https://blogs.plos.org/plos/2019/06/looking-good-tips-for-creating-your-plos-figures-graphics/"

Response: We followed these instructions and modified the Tables accordingly.

Table 1

Table 2

4. As required by our policy on Data Availability, please ensure your manuscript or supplementary information includes the following:

Response: We followed these instructions and prepared a Table listing all the extracted papers and the reasons for our selections. (Supplemental Table 1)

Response: We have compiled a Table detailing the 14 papers we used for our statistical review. (Supplemental Table 2)

Response: These are included in the manuscript as Figure 2 and Table 2.

Response:The following sentence has been added to the Materials and Methods section.

Missing data w

---

## [Decision Letter · Decision Letter 1]

4 Mar 2025

Parathyroid near-infrared autofluorescence differently benefits depending on the surgeon's skill for preventing from hypoparathyroidism after total thyroidectomy: A systematic review and meta-analysis

PONE-D-24-42113R1

Dear Dr. Takahashi,

We’re pleased to inform you that your manuscript has been judged scientifically suitable for publication and will be formally accepted for publication once it meets all outstanding technical requirements.

Kind regards,

Mohamed Khaled Ibrahim, MD

Guest Editor

PLOS ONE

Additional Editor Comments (optional):

Reviewers' comments:

Reviewer's Responses to Questions

**Comments to the Author**

1. If the authors have adequately addressed your comments raised in a previous round of review and you feel that this manuscript is now acceptable for publication, you may indicate that here to bypass the “Comments to the Author” section, enter your conflict of interest statement in the “Confidential to Editor” section, and submit your "Accept" recommendation.

Reviewer #2: All comments have been addressed

2. Is the manuscript technically sound, and do the data support the conclusions?

Reviewer #2: Yes

3. Has the statistical analysis been performed appropriately and rigorously? 

Reviewer #2: Yes

4. Have the authors made all data underlying the findings in their manuscript fully available?

Reviewer #2: Yes

5. Is the manuscript presented in an intelligible fashion and written in standard English?

Reviewer #2: Yes

6. Review Comments to the Author

Reviewer #2: This manuscript was well revised according to the reviewers comments. This manuscript is worthwhile publication now.

7. PLOS authors have the option to publish the peer review history of their article (what does this mean? ). If published, this will include your full peer review and any attached files.

**Do you want your identity to be public for this peer review?** For information about this choice, including consent withdrawal, please see our Privacy Policy .

Reviewer #2: No

---

## [Editor Report · Acceptance letter]

PONE-D-24-42113R1

PLOS ONE

Dear Dr. Takahashi,

I'm pleased to inform you that your manuscript has been deemed suitable for publication in PLOS ONE. Congratulations! Your manuscript is now being handed over to our production team.

Kind regards,

on behalf of

Dr. Mohamed Khaled Ibrahim

Guest Editor

PLOS ONE